# Security Analysis of the Image Encryption Algorithm Based on a Two-Dimensional Infinite Collapse Map

**DOI:** 10.3390/e24081023

**Published:** 2022-07-25

**Authors:** Guanwen Shi, Simin Yu, Qianxue Wang

**Affiliations:** College of Automation, Guangdong University of Technology, Guangzhou 510006, China; sgwxuexi@126.com (G.S.); siminyu@163.com (S.Y.)

**Keywords:** chaotic image encryption, cryptanalysis, 2D infinite collapse map, equivalent key

## Abstract

This paper analyzes the security of the image encryption algorithm based on a two-dimensional (2D) infinite collapse map. The encryption algorithm adopts a permutation–diffusion structure and can perform two or more rounds to achieve a higher level of security. By cryptanalysis, it is found that the original diffusion process can be split into a permutation–diffusion structure, which comes after the original permutation, so these two permutations can be merged into one. Then, some theorems about round-down operation are summarized, and the encryption and decryption equations in the diffusion process are deduced and simplified accordingly. Since the chaotic sequences used in encryption algorithm are independent of the plaintext and ciphertext, there are equivalent keys. The original encryption algorithm with single-round, two-round, and multi-round of permutation–diffusion processes is cracked, and the data complexity of the cryptanalysis attacks is analyzed. Numerical simulation is carried out by MATLAB, and the experimental results and theoretical analysis show the effectiveness of the cryptanalysis attacks. Finally, some suggestions for improvement are given to overcome the shortcomings of the original encryption algorithm.

## 1. Introduction

Advances in information and network technology have facilitated the rapid development of the Internet in providing the technical foundation, and the Internet is deeply integrated into all aspects of human life. Accompanying this is a variety of data forms and massive amounts of data generated every day. Since these data are closely linked with user information, their protection is particularly important. Digital image data are an important carrier of information, and has occupied a large part in the process of network transmission. Encrypting images is an important means to ensure image security.

Image data have the characteristics of strong correlation between pixels, high data redundancy, and large amount of data. The traditional text encryption algorithms such as DES and AES are not suitable for image encryption. In recent years, image encryption based on chaotic systems [1,2,3,4,5,6,7], cellular automata [8,9,10,11,12], DNA encoding [13,14,15], bit plane decomposition [16,17,18,19,20,21], and elliptic curve [22,23,24,25,26] is the mainstream of cryptography. Due to the significant properties of unpredictability, ergodicity and initial state sensitivity, the chaotic system becomes a good choice for encryption [27]. However, the chaotic sequence is transformed to a bit sequence to encrypt the plaintext in most chaotic image encryptions. The security of the encryption is thus determined by the properties of the bit sequence. Moreover, the essential reason for the chaotic cryptosystem easily existing equivalent keys is that the encryption process is independent of plaintext and/or ciphertext. In addition, elliptic curve cryptography is capable of providing high security than to other cryptosystems with the same key size because it is more complicated and requires a deeper mathematical understanding; it is more susceptible to errors which diminishes its security.

Since Matthews proposed a generalized logistic map and used it to generate pseudo-random numbers for data encryption [28], a large number of scholars have poured into using chaotic systems to design and implement novel image encryption schemes. In 1998, Fridrich [29] first proposed a chaotic image encryption scheme with multi-round of permutation–diffusion processes, which gradually became the main operation in chaotic image encryption algorithms. In 2015, Simin Yu reviewed the current situation and existing problems of the theory and application of chaotic cryptography, the literature [30] focused on the progress of high-dimensional chaotic cryptography and its application in multimedia secure communication and hardware implementation technology. In 2018, Özkaynak reported various chaotic image encryption algorithms proposed in the past 20 years. The chaotic systems, diffusion operations, and analysis methods commonly used in chaotic image encryption algorithms are classified and summarized in detail [31]. Overall, the chaotic encryption algorithm with a multi-round of permutation–diffusion processes offers cryptographic properties better than those with a single-round of permutation–diffusion processes, and it can resist against the chosen-plaintext attacks.

Cryptography and cryptanalysis are the unity of opposites, and they promote each other and develop together. Through cryptanalysis, the defects of cryptographic algorithm can be pointed out and the suggestions for improvement are given. Cryptanalysis is based on the Kerchhoff’s principle; a cryptographic system should be secure even if everything about the system, except the key, is public knowledge. The attacker can get the plaintext or even the encryption key through the obtained plaintext/ciphertext pair. In cryptology, the basic models are named after the generally defined attacks such as ciphertext-only attack, known-plaintext attack, chosen-plaintext attack, and chosen-ciphertext attack. Many analysis methods can be classified into the above four methods. In recent years, linear attack and differential attack [32] have been proposed one after another, which have a great impact on cryptanalysis. Many new analysis methods are variants of these two methods [33].

At present, there are some analytical articles on a multi-round image encryption algorithm. In 2010, Solak et al. proposed a chosen-ciphertext attack on the Fridrich’s scheme for the first time [34]. Some bases for further optimizing attack on the Fridrich’s scheme and its variants are provided in [35]. In 2015, Chen et al. analyzed an encryption algorithm with a multi-round of permutation–diffusion structure [36], and proposed a differential cryptanalysis method for two-round and multi-round [37]. However, due to the special permutation operation adopted by the original encryption algorithm, the analysis for multi-round is not universal; in 2016, they proposed a method of chosen-ciphertext attack, and verified the adaptability of this attack method by analyzing several common diffusion equations [38]. In 2021, a multi-round chaotic image encryption algorithm was analyzed in [39]. The original encryption algorithm adopts multiple permutations and one diffusion, and repeats them multiple rounds. Multiple consecutive permutations are equivalent to one permutation. Since the diffusion operation only uses XOR without ciphertext feedback, the diffusion part can be separated from the permutation part. Therefore, it can be cracked by simplifying it into one round of permutation–diffusion. In the same year, Chen et al. mathematically summarized and expressed a class of chaotic image cryptosystems based on a multi-round of permutation–diffusion structure [37,38], and proposed a chosen-ciphertext attack method for this kind of encryption algorithm [40,41]. It is noted that the cryptanalysis algorithms in the existing literature are mainly aimed at the single-round encryption and some multi-round encryptions, which also can be directly equivalent to single-round encryption after simplification.

In this paper, a security analysis of the image encryption algorithm based on a 2D infinite collapse map proposed in [42] is carried out. According to the analysis, the encryption algorithm has one permutation operation in the diffusion process. Therefore, its structure is actually a permutation–permutation–diffusion structure, and two permutation operations can be equivalent to one permutation operation. In addition, this paper deduces the rules of round-down operation, and then gives the correct diffusion decryption equation. Since the chaotic sequences used in the encryption algorithm are independent of the plaintext and ciphertext, there are equivalent keys. This paper analyzes and discusses the single-round, two-round and multi-round situations, provides the attack complexity, and gives the corresponding improvement suggestions to overcome the shortcomings of the original encryption algorithm. The main advantage of this paper is that a detailed security analysis of a more complex multi-round encryption algorithm is carried out, and the main difference between this multi-round encryption and the previous multi-round encryption methods is that the multi-round encryption cannot be directly equivalent to a single-round of encryption. Therefore, the cryptanalysis methods in the existing literature cannot be directly used to crack this multi-round encryption algorithm.

The remainder of this article is organized as follows: Some definitions and related theorems are provided in Section 2. Section 3 presents the detail of the original encryption algorithm, and gives the correct decryption equation. An analysis of the encryption algorithm is demonstrated in detail in Section 4. Section 5 mainly introduces the numerical simulation experiments carried out by MATLAB. The experimental results verify the correctness of the cryptanalysis, and at the same time, the complexity of the deciphering algorithms is given, and corresponding improvement measures are proposed to overcome the shortcomings of the original encryption algorithm. The last section concludes the article.

## 2. Some Definitions and Related Theorems

In order to better analyze the original encryption algorithm, it is first necessary to simplify the original algorithm. According to the formula used in the original algorithm, some preliminaries are given to aid the subsequent theoretical analysis. The definitions and properties of round-down operation ⌊·⌋, the operation {·} for finding the fractional part of a real number, and the modulus operator are introduced, and three theorems about these operations are deduced in this section.

**Definition** **1**([43]). *The largest integer of a real number a is recorded as ⌊a⌋, which is the largest integer less than or equal to a, that is, ⌊a⌋ is the integer satisfying ⌊a⌋≤a<⌊a⌋+1.*

**Definition** **2**([43]). *The fractional part of the real number a is recorded as {a}, which is the difference between a and ⌊a⌋, that is, {a}=a−⌊a⌋.*

**Property** **1.**
*a=⌊a⌋+{a},0≤{a}<1.*


**Property** **2.**
*⌊n+a⌋=n+⌊a⌋,{n+a}={a},n∈Z.*


**Theorem** **1.**
*For any real numbers a and b, there are*

$$\left\lfloor a+b\right\rfloor =\left\{\begin{array}{cc} \left\lfloor a\rfloor +\lfloor b\right\rfloor & 0\leq \{a\}+\{b\}<1,\\ \lfloor a\rfloor +\lfloor b\rfloor +1 & 1\leq \{a\}+\{b\}<2. \end{array}\right.$$



**Proof.** 

$$\begin{array}{ccc} \left\lfloor a+b\right\rfloor & =\lfloor (\lfloor a\rfloor +\{a\})+(\lfloor b\rfloor +\{b\})\rfloor & \left(\mathrm{Property 1}\right)\\ \; & =\lfloor \lfloor a\rfloor +\lfloor b\rfloor +(\{a\}+\{b\})\rfloor & \\ \; & =\lfloor a\rfloor +\lfloor b\rfloor +\lfloor \{a\}+\{b\}\rfloor . & (\lfloor a\rfloor +\lfloor b\rfloor \in Z,\mathrm{Property 2}). \end{array}$$

From Property 1, we know 0≤{a}<1,0≤{b}<1, so 0≤{a}+{b}<2.When 0≤{a}+{b}<1, ⌊{a}+{b}⌋=0, then ⌊a+b⌋=⌊a⌋+⌊b⌋.When 1≤{a}+{b}<2, ⌊{a}+{b}⌋=1, then ⌊a+b⌋=⌊a⌋+⌊b⌋+1.
□

**Theorem** **2.**
*For any real numbers a and b, there are*

$$\left\lfloor a-b\right\rfloor =\left\{\begin{array}{cc} \left\lfloor a\rfloor -\lfloor b\right\rfloor & 0\leq \{a\}-\{b\}<1,\\ \lfloor a\rfloor -\lfloor b\rfloor -1 & -1<\{a\}-\{b\}<0. \end{array}\right.$$



**Proof.** 

$$\begin{array}{ccc} \left\lfloor a-b\right\rfloor & =\lfloor (\lfloor a\rfloor +\{a\})-(\lfloor b\rfloor +\{b\})\rfloor & \left(\mathrm{Property 1}\right)\\ \; & =\lfloor \lfloor a\rfloor -\lfloor b\rfloor +(\{a\}-\{b\})\rfloor & \\ \; & =\lfloor a\rfloor -\lfloor b\rfloor +\lfloor \{a\}-\{b\}\rfloor . & \left(\lfloor a\rfloor -\lfloor b\rfloor \in Z,\mathrm{Property 2}\right) \end{array}$$

From Property 1, we know 0≤{a}<1,0≤{b}<1, so −1<{a}−{b}<1.When 0≤{a}−{b}<1, ⌊{a}−{b}⌋=0, then ⌊a−b⌋=⌊a⌋−⌊b⌋.When −1<{a}+{b}<0, ⌊{a}−{b}⌋=−1, then ⌊a−b⌋=⌊a⌋−⌊b⌋−1.
□

**Definition** **3**([44]). *The modular operation returns the remainder after a real number is divided by a positive integer, and often abbreviated as mod:*

**Property** **3.**

(amod256)mod256=amod256,a∈R.



**Property** **4.**

(a+b)mod256=((amod256)+(bmod256))mod256,a,b∈R.



**Theorem** **3.**

⌊amod256⌋=⌊a⌋mod256,a∈R.



**Proof.** Assuming b=amod256, the corresponding inverse operation is a=256×k+b, where a,b∈R, k∈Z and 0≤b<256, so ⌊amod256⌋=⌊b⌋ and
$$\begin{array}{cccc} \lfloor a\rfloor \;\mathrm{mod}\;256 & =\lfloor 256\times k+b\rfloor \;\mathrm{mod}\;256 & \\ \; & =(256\times k+\lfloor b\rfloor )\;\mathrm{mod}\;256 & \; & \left(256\times k\in Z,Property\;2\right)\\ \; & =\lfloor b\rfloor . & \; & \left(0\leq \lfloor b\rfloor <256,\mathrm{Definition 3}\right) \end{array}$$⌊amod256⌋=⌊b⌋=⌊a⌋mod256 is proved. □

## 3. Description of the Original Encryption Algorithm

In this section, the chaotic map used in [42] is first introduced, and then the original encryption algorithm is described in detail.

### 3.1. Two-Dimensional Infinite Collapse Map (2D-ICM)

The chaotic system 2D-ICM used in the original encryption algorithm is a two-dimensional infinite collapse map obtained by integrating two one-dimensional infinite collapse maps with different parameters [42], and its iterative equation is
(1)$$\begin{array}{c} \left\{\begin{array}{c} x_{n+1}=\sin\left(\frac{a}{y_{n}}\right)\cdot \sin\left(\frac{b}{x_{n}}\right),\\ y_{n+1}=\sin\left(\frac{a}{x_{n}}\right)\cdot \sin\left(\frac{b}{y_{n}}\right), \end{array}\right. \end{array}$$
where the control parameters *a* and *b* are real numbers, a≠0, b≠0, and the initial states are recorded as $x_{0},y_{0}$.

### 3.2. 2D-ICM Based Image Encryption Algorithm (ICMIE)

According to [42], it proposed a new image encryption algorithm based on 2D-ICM and named it ICMIE. The original algorithm ICMIE is described as follows:

(1)Key parameters

There are seven key parameters in the original algorithm. The key *K* is expressed as $a_{0},b_{0},x,y,T,C_{1},C_{2}$, and the first five parameters $a_{0},b_{0},x,y,T$ are 40-bit binary representation. Assuming that the 40-bit binary is ${\left(s_{1}s_{2}\cdots s_{40}\right)}_{2}$, the IEEE754 format
(2)$$\begin{array}{c} d=\frac{\sum_{i=1}^{40}s_{i}2^{40-i}}{2^{40}} \end{array}$$
is adopted to convert $a_{0},b_{0},x,y,T$ into decimal numbers in $\left[0,1\right)$, then $C_{1}$ and $C_{2}$ are positive integers represented by 20-bit binary, and they are converted into decimal numbers directly. Substitute the converted decimal $a_{0},b_{0},x,y,T,C_{1},C_{2}$ into the following equation:(3)$$\begin{array}{c} \left\{\begin{array}{c} a=\left(a_{0}+T\times C_{1}\right)\;\mathrm{mod}\;5+16,\\ b=\left(b_{0}+T\times C_{2}\right)\;\mathrm{mod}\;5+16,\\ x_{0}=\left(x+T\times C_{1}\right)\;\mathrm{mod}\;2-1,\\ y_{0}=\left(y+T\times C_{2}\right)\;\mathrm{mod}\;2-1. \end{array}\right. \end{array}$$

The initial conditions $a,b,x_{0},y_{0}$ of 2D-ICM can be obtained.

(2)Encryption process

The original algorithm divides the encryption process into permutation and diffusion, and then performs two or more rounds of permutation and diffusion as a whole. In fact, the diffusion process of the original algorithm also includes a permutation operation. In order to distinguish, the first permutation operation is named permutation 1 and the second permutation operation is named permutation 2. The grayscale image *P* of M×N is encrypted, and the ciphertext image *C* of the same size is finally generated. The overall encryption process is shown in Figure 1.

The specific encryption steps are described as follows:

(1) Permutation

First, two chaotic matrices *X* and *Y* of M×N are generated by 2D-ICM. The matrix *S* is combined into a single matrix S=X∗Y by multiplying the corresponding elements of *X* and *Y*. The index matrix *I* is composed of the position of each element in the original matrix *S* after sorting *S* in ascending order. Then, the pixel positions of the plaintext image *P* are rearranged by using the index matrix *I* to obtain the permutation 1 image *F*.

(2) Diffusion

First, the index matrix χ is composed of the corresponding positions of all elements of the chaotic matrix *X* in its ascending order. Then, the pixel positions of the permutation 1 image *F* are arranged again by the index matrix χ to obtain a new permutation 2 image *A*. Finally, the pixel values of the diffusion image *D* are obtained by the following method:(4)$$d_{i}=\left\{\begin{array}{cc} \left\lfloor \left(a_{i}+a_{M\times N}+\left|y_{i}\right|\times \left(2^{31}-1\right)\right)\;\mathrm{mod}\;256\right\rfloor & \;\mathrm{if}\;i=1,\\ \left\lfloor \left(a_{i}+d_{i-1}+\left|y_{i}\right|\times \left(2^{31}-1\right)\right)\;\mathrm{mod}\;256\right\rfloor & \;\mathrm{if}\;i\in \left[2,M\times N\right], \end{array}\right.$$
where i=1,2,…,M×N, then $d_{i},a_{i}$ and $y_{i}$ are the pixel values of the *i*-th element of the diffusion image *D*, the permutation 2 image *A* and the chaotic matrix *Y* according to the raster scan order, respectively.

(3) Repeat steps (1) and (2) to achieve multi-round encryption.

(3)Decryption process

Usually, the decryption process is the inverse of the image encryption process. Using the correct key to generate the chaotic matrices *X* and *Y*, the decryption process of ICMIE will alternately perform the inverse diffusion and inverse permutation in two rounds or multiple rounds, and then obtain the recovered image. The decryption equation in the diffusion process is incorrect. When $i\in \left[2,M\times N\right]$, according to the encryption Equation (Equation 4),
$$\begin{array}{ccc} d_{i}\;\; & =\left\lfloor \left(a_{i}+d_{i-1}+\left|y_{i}\right|\times \left(2^{31}-1\right)\right)\;\mathrm{mod}\;256\right\rfloor , & \\ d_{i}\;\; & =\left\lfloor \left(a_{i}+d_{i-1}+\left|y_{i}\right|\times \left(2^{31}-1\right)\right)\right\rfloor \;\mathrm{mod}\;256, & (\mathrm{Theorem 3})\\ d_{i}+256\times k_{i} & =\left\lfloor \left(a_{i}+d_{i-1}+\left|y_{i}\right|\times \left(2^{31}-1\right)\right)\right\rfloor , & (\mathrm{Definition 3})\\ d_{i}+256\times k_{i} & =a_{i}+d_{i-1}+\left\lfloor \left(\left|y_{i}\right|\times \left(2^{31}-1\right)\right)\right\rfloor , & (\mathrm{Property 2})\\ a_{i}\;\; & =d_{i}+256\times k_{i}-d_{i-1}-\left\lfloor \left(\left|y_{i}\right|\times \left(2^{31}-1\right)\right)\right\rfloor , & \\ a_{i}\;\mathrm{mod}\;256\; & =\left(d_{i}+256\times k_{i}-d_{i-1}-\left\lfloor \left(\left|y_{i}\right|\times \left(2^{31}-1\right)\right)\right\rfloor )\;\mathrm{mod}\;256\right., & \\ a_{i}\;\; & =\left(\left(d_{i}-d_{i-1}-\left\lfloor \left(\left|y_{i}\right|\times \left(2^{31}-1\right)\right)\right\rfloor \right)\;\mathrm{mod}\;256\right)\;\mathrm{mod}\;256, & (\mathrm{Property 4})\\ a_{i}\;\; & =\left(d_{i}-d_{i-1}-\left\lfloor \left(\left|y_{i}\right|\times \left(2^{31}-1\right)\right)\right\rfloor \right)\;\mathrm{mod}\;256, & (\mathrm{Property 3})\\ a_{i}\;\; & =\left\lfloor d_{i}-d_{i-1}-\left(\left|y_{i}\right|\times \left(2^{31}-1\right)\right)+1\right\rfloor \;\mathrm{mod}\;256, & (\mathrm{Theorem 2}) \end{array}$$
where $k_{i}\in Z\left(i=2,3,\cdots ,M\times N\right)$. Similarly, $a_{i}$ can be obtained when i=1, and the correct decryption equation is finally derived as
(5)$$a_{i}=\left\{\begin{array}{cc} \left\lfloor \left(d_{i}-d_{i-1}-\left(\left|y_{i}\right|\times \left(2^{31}-1\right)\right)+1\right)\;\mathrm{mod}\;256\right\rfloor & \;\mathrm{if}\;i\in \left[2,M\times N\right],\\ \left\lfloor \left(d_{i}-a_{M\times N}-\left(\left|y_{i}\right|\times \left(2^{31}-1\right)\right)+1\right)\;\mathrm{mod}\;256\right\rfloor & \;\mathrm{if}\;i=1. \end{array}\right.$$

Then, the pixel positions of the ciphertext image will be processed by inverse permutation. The original image is completely recovered.

## 4. Cryptanalysis

The generation process of the chaotic sequences and the encryption process of the original algorithm are independent of the plaintext and the ciphertext, so there are equivalent keys. Firstly, the core structure of the original encryption algorithm (i.e., the permutation–permutation–diffusion structure) is generalized, then the diffusion equation is isolated, and the two permutation processes are merged into one permutation. Next, the original encryption algorithm is analyzed in terms of single-round, two-round, and multi-round.

The original encryption algorithm can be summarized as a multi-round of permutation–diffusion processes as shown in Figure 2.

In Figure 2, KP and KD represent the permutation equivalent key and the diffusion equivalent key, respectively. *P* represents plaintext image and *C* represents ciphertext image. *n* represents the total number of rounds of encryption, and
$$P^{\left(t\right)}=\left\{\begin{array}{cc} P & t=1,\\ D^{\left(t-1\right)} & t\in [2,n], \end{array}\right.$$
where $P^{\left(t\right)}$, $A^{\left(t\right)}$, and $D^{\left(t\right)}$ represent the plaintext image, the permutation image, and diffusion image encrypted in the *t*-th round, respectively. Taking the feedback apart, it can be shown in Figure 3.

In Figure 3, $P^{\left(t\right)}=D^{\left(t-1\right)}$, that is, the diffusion image encrypted in the previous round is the plaintext image encrypted in the subsequent round. For plaintext image *P* and ciphertext image *C*, there are $P^{\left(1\right)}=P$ and $C=D^{\left(n\right)}$.

For the convenience of the following discussion, some definitions are given here. $P_{i}^{\left(t\right)},A_{i}^{\left(t\right)},D_{i}^{\left(t\right)}$ respectively represent the *i*-th plaintext image, permutation image, and diffusion image in the *t*-th round. The size of the images discussed in this paper are all M×N.

### 4.1. Simplification of ICMIE

Since the two permutations are independent of plaintext, they can be equivalent to one permutation operation. The equivalent key KP of the two permutation operations from plaintext image *P* to permutation image *A* can be obtained directly in one step. The diffusion equivalent key KD can be obtained from the permutation image *A* to the diffusion image *D*.

The original diffusion encryption Equation (Equation 4) is deduced and the following equation is obtained. When $i\in \left[2,M\times N\right]$, according to Theorem 1, there is
(6)$$\begin{array}{ccc} d_{i} & =\left\lfloor \left(a_{i}+d_{i-1}+\left|y_{i}\right|\times \left(2^{31}-1\right)\right)\;\mathrm{mod}\;256\right\rfloor \\ \; & =\left\lfloor a_{i}+d_{i-1}+\left|y_{i}\right|\times \left(2^{31}-1\right)\right\rfloor \;\mathrm{mod}\;256 & (\mathrm{Theorem 3})\\ \; & =\left(a_{i}+d_{i-1}+\left\lfloor \left|y_{i}\right|\times \left(2^{31}-1\right)\right\rfloor \right)\;\mathrm{mod}\;256 & (\mathrm{Property 2})\\ \; & =\left(a_{i}+d_{i-1}+\left\lfloor \left|y_{i}\right|\times \left(2^{31}-1\right)\right\rfloor \;\mathrm{mod}\;256\right)\;\mathrm{mod}\;256 & (\mathrm{Property 4})\\ \; & =\left(a_{i}+d_{i-1}+{\hat{y}}_{i}\right)\;\mathrm{mod}\;256, \end{array}$$
where ${\hat{y}}_{i}=\left(\left\lfloor \left|y_{i}\right|\times \left(2^{31}-1\right)\right\rfloor \right)\;\mathrm{mod}\;256$.

Likewise, when i=1,
(7)$$d_{1}=\left(a_{1}+a_{M\times N}+{\hat{y}}_{1}\right)\;\mathrm{mod}\;256,$$
where ${\hat{y}}_{1}=\left(\left\lfloor \left|y_{1}\right|\times \left(2^{31}-1\right)\right\rfloor \right)\;\mathrm{mod}\;256$.

In order to facilitate the analysis, ∔ is defined to represent the modular addition that is, the two elements are added and then modulo 256. Correspondingly, $\dot{-}$ represents the modular subtraction, that is, the two elements are subtracted and then modulo 256. From Equations (Equation 6) and (Equation 7),
(8)$$\begin{array}{c} d_{i}=\left\{\begin{array}{cc} a_{i}∔a_{M\times N}∔{\hat{y}}_{i} & \;\mathrm{if}\;i=1,\\ a_{i}∔d_{i-1}∔{\hat{y}}_{i} & \;\mathrm{if}\;i\in \left[2,M\times N\right], \end{array}\right. \end{array}$$
where ${\hat{y}}_{i}=\left(\left\lfloor \left|y_{i}\right|\times \left(2^{31}-1\right)\right\rfloor \right)\;\mathrm{mod}\;256$.

### 4.2. Security Analysis of Encryption in Single-Round

First, let $P_{0}$ be an all-zero image, then the pixel value will not be changed after permutation. Therefore, the element values of the permutation image $A_{0}$ are all 0. The image $D_{0}$ is obtained according to the encryption algorithm. According to Equation (Equation 8), one has
(9)$$\begin{array}{c} d_{i}=\left\{\begin{array}{cc} {\hat{y}}_{i} & \;\mathrm{if}\;i=1,\\ d_{i-1}∔{\hat{y}}_{i} & \;\mathrm{if}\;i\in \left[2,M\times N\right], \end{array}\right. \end{array}$$
and
(10)$$\begin{array}{c} {\hat{y}}_{i}=\left\{\begin{array}{cc} d_{i} & \;\mathrm{if}\;i=1,\\ d_{i}-d_{i-1}+256k_{i} & \;\mathrm{if}\;i\in \left[2,M\times N\right], \end{array}\right. \end{array}$$
where $k_{i}\in Z\left(i=2,3,\cdots ,M\times N\right)$.

Because ${\hat{y}}_{i}$ and $d_{i}$ perform modulo 256 operation,

${\hat{y}}_{i}\in \left\{0,1,2,\cdots ,255\right\}$,$\;d_{i}-d_{i-1}\in \left\{-255,-254,\cdots ,254,255\right\}\left(i=2,3,\cdots ,M\times N\right)$, so
(11)$$\begin{array}{c} {\hat{y}}_{i}=\left\{\begin{array}{cc} d_{i} & \;\mathrm{if}\;i=1,\\ d_{i}-d_{i-1}+256 & \;\mathrm{if}\;i\in \left[2,M\times N\right]\mathrm{and}\left(d_{i}-d_{i-1}\right)<0,\\ d_{i}-d_{i-1} & \;\mathrm{if}\;i\in \left[2,M\times N\right]\mathrm{and}\left(d_{i}-d_{i-1}\right)\geq 0. \end{array}\right. \end{array}$$

In other words, substitute the pixel value of $D_{0}$ to obtain ${\hat{y}}_{i}$ by Equation (Equation 11), then make $kd_{i}={\hat{y}}_{i}\left(i=1,2,\cdots ,M\times N\right)$ to get the equivalent key $KD=kd_{1}kd_{2}\cdots kd_{M\times N}$. Next, according to Theorem 3, the diffusion decryption from Equation (Equation 5), one can further obtain
(12)$$\begin{array}{cc} a_{i} & =\left\{\begin{array}{cc} \left\lfloor d_{i}\dot{-}d_{i-1}\dot{-}\left(\left|y_{i}\right|\times \left(2^{31}-1\right)\right)∔1\right\rfloor & \;\mathrm{if}\;i\in \left[2,M\times N\right],\\ \left\lfloor d_{i}\dot{-}a_{M\times N}\dot{-}\left(\left|y_{i}\right|\times \left(2^{31}-1\right)\right)∔1\right\rfloor & \;\mathrm{if}\;i=1. \end{array}\right.\\ \; & =\left\{\begin{array}{cc} \left\lfloor \left(d_{i}\dot{-}d_{i-1}∔1\right)\dot{-}\left(\left|y_{i}\right|\times \left(2^{31}-1\right)\right)\right\rfloor & \;\mathrm{if}\;i\in \left[2,M\times N\right],\\ \left\lfloor \left(d_{i}\dot{-}a_{M\times N}∔1\right)\dot{-}\left(\left|y_{i}\right|\times \left(2^{31}-1\right)\right)\right\rfloor & \;\mathrm{if}\;i=1. \end{array}\right. \end{array}$$

Because $d_{i}\dot{-}d_{i-1}∔1,d_{i}\dot{-}a_{M\times N}∔1\in Z$, it means $\left\{d_{i}\dot{-}d_{i-1}∔1\right\}=0$ and $\left\{d_{i}\dot{-}a_{M\times N}∔1\right\}=0$, then $\left\{d_{i}\dot{-}d_{i-1}∔1\right\}\dot{-}\left\{\left|y_{i}\right|\times \left(2^{31}-1\right)\right\}<0$ and $\left\{d_{i}\dot{-}a_{M\times N}∔1\right\}\dot{-}\left\{\left|y_{i}\right|\times \left(2^{31}-1\right)\right\}<0$. According to Theorem 2 and Property 2,
(13)$$\begin{array}{cc} a_{i} & =\left\{\begin{array}{cc} \left\lfloor d_{i}\dot{-}d_{i-1}∔1\right\rfloor \dot{-}\left\lfloor \left|y_{i}\right|\times \left(2^{31}-1\right)\right\rfloor \dot{-}1 & \;\mathrm{if}\;i\in \left[2,M\times N\right],\\ \left\lfloor d_{i}\dot{-}a_{M\times N}∔1\right\rfloor \dot{-}\left\lfloor \left|y_{i}\right|\times \left(2^{31}-1\right)\right\rfloor \dot{-}1 & \;\mathrm{if}\;i=1, \end{array}\right.\\ \; & =\left\{\begin{array}{cc} d_{i}\dot{-}d_{i-1}∔1\dot{-}\left\lfloor \left|y_{i}\right|\times \left(2^{31}-1\right)\right\rfloor \dot{-}1 & \;\mathrm{if}\;i\in \left[2,M\times N\right],\\ d_{i}\dot{-}a_{M\times N}∔1\dot{-}\left\lfloor \left|y_{i}\right|\times \left(2^{31}-1\right)\right\rfloor \dot{-}1 & \;\mathrm{if}\;i=1, \end{array}\right.\\ \; & =\left\{\begin{array}{cc} d_{i}\dot{-}d_{i-1}\dot{-}{\hat{y}}_{i} & \;\mathrm{if}\;i\in \left[2,M\times N\right],\\ d_{i}\dot{-}a_{M\times N}\dot{-}{\hat{y}}_{i} & \;\mathrm{if}\;i=1, \end{array}\right. \end{array}$$
where ${\hat{y}}_{i}=\left(\left\lfloor \left|y_{i}\right|\times \left(2^{31}-1\right)\right\rfloor \right)\;\mathrm{mod}\;256$.

The permutation image *A* corresponding to the ciphertext image *C* can be cracked by substituting the equivalent key KD (i.e., ${\hat{y}}_{i}=kd_{i}\left(i=1,2,\cdots ,M\times N\right)$) obtained from all-zero plaintext and $d_{i}\left(i=1,2,\cdots ,M\times N\right)$ according to Equation (Equation 13). Because the specific ciphertext *C* is known, then the diffusion image D=C, so $d_{i}\left(i=1,2,\cdots ,M\times N\right)$ is known.

Since the two permutations are independent of the plaintext, they can be equivalent to one permutation, and the permutation operation only changes the coordinate position of the pixel without changing the pixel value, so that only the coordinate position of the pixel in the permutation image *A* is changed. Therefore, the equivalent permutation key KP can be solved by comparing the pixel pairs of the plaintext images and the permutation images. Next, the optimal chosen-plaintext attack is used [45], and the steps are as follows:

Step 1: Construct a data matrix *U* with the same size as the image *P*, $u_{j}$ is the value of the *j*-th element of the matrix *U* obtained in raster scan order. The nonnegative integers 0,1,⋯,M×N−1 are successively written into the data matrix *U* according to the raster scan order by $u_{j}=j-1\left(j=1,2,3,\ldots ,M\times N\right)$.

Step 2: Calculate the number of selected plaintexts $l=\left\lceil \log_{256}\left(M\times N\right)\right\rceil$, where $\left\lceil \cdot \right\rceil$ is the round-up operation. In addition, create *l* plaintext images $P_{1},P_{2},\cdots ,P_{l}$.

Step 3: Use *U* to write the value into $P_{1},P_{2},\cdots ,P_{l}$. The writing rule of the *j*-th element $p_{i,j}$ obtained from the *i*-th plaintext image in raster scan order is
(14)$$p_{i,j}=\left\lfloor \left(u_{j}/256^{i-1}\right)\%256\right\rfloor ,$$
where i=1,2,3,…,l and j=1,2,3,…,M×N.

After constructing the plaintext through the above steps, *l* plaintext images $P_{1},P_{2},\cdots ,P_{l}$ are successively input into the encryptor to obtain the corresponding ciphertext images $C_{1},C_{2},\cdots ,C_{l}$, respectively. Then, according to the obtained equivalent diffusion key KD, inverse diffusion is carried out to obtain $A_{1},A_{2},\cdots ,A_{l}$, respectively, and these images are combined into a data matrix *V*. The consolidation rule is
(15)$$V=\sum_{i=1}^{l}\left(A_{i}\times 256^{i-1}\right),$$
where i=1,2,3,…,l. By comparing the position difference between the data matrix *U* and the data matrix *V* with the same pixel value, the equivalent permutation key KP used in permutation can be obtained.

### 4.3. Cryptanalysis of Two-Round Encryption

Two-round encryption is analyzed here by the combination of the differential attack and the chosen-plaintext attack.

Firstly, the encryption algorithm is deduced by differential analysis. According to Equation (Equation 8),
(16)$$\begin{array}{c} \left\{\begin{array}{cc} d_{1} & =a_{M\times N}∔a_{1}∔{\hat{y}}_{1},\\ d_{2} & =a_{M\times N}∔a_{1}∔a_{2}∔{\hat{y}}_{1}∔{\hat{y}}_{2},\\ \; & \vdots \\ d_{j} & =a_{M\times N}∔a_{1}∔a_{2}∔\ldots ∔a_{j}∔{\hat{y}}_{1}∔{\hat{y}}_{2}∔\ldots ∔{\hat{y}}_{j},\\ \; & \vdots \\ d_{M\times N} & =a_{M\times N}∔a_{1}∔a_{2}∔\ldots ∔a_{M\times N}∔{\hat{y}}_{1}∔{\hat{y}}_{2}∔\ldots ∔{\hat{y}}_{M\times N}. \end{array}\right. \end{array}$$

Now use $a_{i,j}^{\left(t\right)},d_{i,j}^{\left(t\right)}$(t=1,2,3,…,n, i=0,1,2,⋯ and j=1,2,3,…,M×N) to represent the *j*-th element of the *i*-th permutation image and diffusion image in the raster scan order in the *t*-th round of encryption, respectively. Then, the *j*-th element in raster scan order in two different diffusion images $D_{k}^{\left(t\right)}$ and $D_{l}^{\left(t\right)}$ encrypted in the *t*-th round can be expressed as
(17)$$d_{k,j}^{\left(t\right)}=a_{k,M\times N}^{\left(t\right)}∔a_{k,1}^{\left(t\right)}∔a_{k,2}^{\left(t\right)}∔\ldots ∔a_{k,j}^{\left(t\right)}∔{\hat{y}}_{1}∔{\hat{y}}_{2}∔\ldots ∔{\hat{y}}_{j}$$
and
(18)$$d_{l,j}^{\left(t\right)}=a_{l,M\times N}^{\left(t\right)}∔a_{l,1}^{\left(t\right)}∔a_{l,2}^{\left(t\right)}∔\ldots ∔a_{l,j}^{\left(t\right)}∔{\hat{y}}_{1}∔{\hat{y}}_{2}∔\ldots ∔{\hat{y}}_{j}$$

Let $\Delta D_{k-l}^{\left(t\right)}=D_{k}^{\left(t\right)}\dot{-}D_{l}^{\left(t\right)}$, the difference $\Delta d_{k-l,j}^{\left(t\right)}=d_{k,j}^{\left(t\right)}\dot{-}d_{l,j}^{\left(t\right)}$ of the *j*-th element of the *t*-th round of diffusion images $D_{k}^{\left(t\right)}$ and $D_{l}^{\left(t\right)}$ in the raster scan order can be obtained, which is
(19)$$\Delta d_{k-l,j}^{\left(t\right)}=\left(a_{k,M\times N}^{\left(t\right)}∔a_{k,1}^{\left(t\right)}∔a_{k,2}^{\left(t\right)}∔\ldots ∔a_{k,j}^{\left(t\right)}\right)\dot{-}\left(a_{l,M\times N}^{\left(t\right)}∔a_{l,1}^{\left(t\right)}∔a_{l,2}^{\left(t\right)}∔\ldots ∔a_{l,j}^{\left(t\right)}\right).$$

Let
$$\left\{\begin{array}{cc} \Delta a_{k-l,M\times N}^{\left(t\right)} & =a_{k,M\times N}^{\left(t\right)}\dot{-}a_{l,M\times N}^{\left(t\right)},\\ \Delta a_{k-l,1}^{\left(t\right)} & =a_{k,1}^{\left(t\right)}\dot{-}a_{l,1}^{\left(t\right)},\\ & \vdots \\ \Delta a_{k-l,j-1}^{\left(t\right)} & =a_{k,j-1}^{\left(t\right)}\dot{-}a_{l,j-1}^{\left(t\right)},\\ \Delta a_{k-l,j}^{\left(t\right)} & =a_{k,j}^{\left(t\right)}\dot{-}a_{l,j}^{\left(t\right)}, \end{array}\right.$$
and there is
(20)$$\Delta d_{k-l,j}^{\left(t\right)}=\Delta a_{k-l,M\times N}^{\left(t\right)}∔\Delta a_{k-l,1}^{\left(t\right)}∔\Delta a_{k-l,2}^{\left(t\right)}∔\ldots ∔\Delta a_{k-l,j}^{\left(t\right)}.$$

It can be seen from the previous analysis that $P^{\left(t\right)}=D^{\left(t-1\right)}$, so there is $\Delta P_{k-l}^{\left(t\right)}=\Delta D_{k-l}^{\left(t-1\right)}$ and $\Delta A_{k-l}^{\left(t\right)}=f_{KP}\left(\Delta P_{k-l}^{\left(t\right)}\right)=f_{KP}\left(\Delta D_{k-l}^{\left(t-1\right)}\right)$, where $f_{KP}\left(\cdot \right)$ is the permutation operation on the matrix. Now, let $a_{i,j}^{\left(t\right)}=f_{kp_{h}}\left(p_{i,h}^{\left(t\right)}\right)$, where j=1,2,…,M×N,h=1,2,…,M×N, and then, $j=kp_{h}$ and $h=kp_{j}^{-1}$ are permutation pairs. $a_{i,j}^{\left(t\right)}=f_{kp_{h}}\left(p_{i,h}^{\left(t\right)}\right)$ indicates that the $j=kp_{h}$-th element $a_{i,j}^{\left(t\right)}$ of the *i*-th permutation image $A_{i}^{\left(t\right)}$ encrypted in the *t*-th round according to the raster scan order is replaced by the $h=kp_{j}^{-1}$-th element $p_{i,h}^{\left(t\right)}$ of the *i*-th plaintext image $P_{i}^{\left(t\right)}$ encrypted in the *t*-th round according to the raster scan order. Thus, the difference of the *j*-th element between the permutation image $A_{k}^{\left(t\right)}$ and $A_{l}^{\left(t\right)}$ encrypted in the *t*-th round is
(21)$$\begin{array}{cc} \Delta a_{k-l,j}^{\left(t\right)} & =a_{k,j}^{\left(t\right)}\dot{-}a_{l,j}^{\left(t\right)}\\ \; & =f_{kp_{h}}\left(p_{k,h}^{\left(t\right)}\right)\dot{-}f_{kp_{h}}\left(p_{l,h}^{\left(t\right)}\right)\\ \; & =f_{kp_{h}}\left(p_{k,h}^{\left(t\right)}\dot{-}p_{l,h}^{\left(t\right)}\right)\\ \; & =f_{kp_{h}}\left(\Delta p_{k-l,h}^{\left(t\right)}\right)\\ \; & =f_{kp_{h}}\left(\Delta d_{k-l,h}^{\left(t-1\right)}\right), \end{array}$$
where $\Delta p_{k-l,h}^{\left(t\right)}=p_{k,h}^{\left(t\right)}\dot{-}p_{l,h}^{\left(t\right)}$.

The flow chart for cracking the two-round encryption is shown in Figure 4. The following is a detailed introduction to the two-round encryption cracking algorithm. It should be pointed out that this method is only for the case of two-round encryption with the same permutation matrix.

Step 1: Construct an all-zero plaintext image as $P_{0}$ of M×N for cracking the ciphertext image *C* of M×N, then construct a plaintext image set $\left\{P_{1},P_{2},\ldots ,P_{M\times N}\right\}$, let the *k*-th element in $P_{k}\left(k\in \left\{1,2,\ldots ,M\times N\right\}\right)$ according to the raster scan order be 1 and the rest be 0, and this means

$P_{1}=\left[\begin{array}{cccc} 1 & 0 & \cdots & 0\\ 0 & 0 & \cdots & 0\\ \vdots & \vdots & & \vdots \\ 0 & 0 & \cdots & 0 \end{array}\right]$, $P_{2}=\left[\begin{array}{cccc} 0 & 1 & \cdots & 0\\ 0 & 0 & \cdots & 0\\ \vdots & \vdots & & \vdots \\ 0 & 0 & \cdots & 0 \end{array}\right]$, ⋯, $P_{M\times N}=\left[\begin{array}{cccc} 0 & 0 & \cdots & 0\\ 0 & 0 & \cdots & 0\\ \vdots & \vdots & & \vdots \\ 0 & 0 & \cdots & 1 \end{array}\right].$

Step 2: According to Equation (Equation 20), the difference relationship between the diffusion images $D_{k}^{\left(2\right)}$ and $D_{0}^{\left(2\right)}$ from their M×N-th to the first element encrypted in the second round is
(22)$$\begin{array}{c} \left\{\begin{array}{cc} \Delta d_{k,M\times N}^{\left(2\right)} & =\Delta a_{k,M\times N}^{\left(2\right)}∔\Delta a_{k,1}^{\left(2\right)}∔\Delta a_{k,2}^{\left(2\right)}∔\ldots ∔\Delta a_{k,M\times N-2}^{\left(2\right)}∔\Delta a_{k,M\times N-1}^{\left(2\right)}∔\Delta a_{k,M\times N}^{\left(2\right)},\\ \Delta d_{k,M\times N-1}^{\left(2\right)} & =\Delta a_{k,M\times N}^{\left(2\right)}∔\Delta a_{k,1}^{\left(2\right)}∔\Delta a_{k,2}^{\left(2\right)}∔\ldots ∔\Delta a_{k,M\times N-2}^{\left(2\right)}∔\Delta a_{k,M\times N-1}^{\left(2\right)},\\ \Delta d_{k,M\times N-2}^{\left(2\right)} & =\Delta a_{k,M\times N}^{\left(2\right)}∔\Delta a_{k,1}^{\left(2\right)}∔\Delta a_{k,2}^{\left(2\right)}∔\ldots ∔\Delta a_{k,M\times N-2}^{\left(2\right)},\\ \; & \vdots \\ \Delta d_{k,2}^{\left(2\right)} & =\Delta a_{k,M\times N}^{\left(2\right)}∔\Delta a_{k,1}^{\left(2\right)}∔\Delta a_{k,2}^{\left(2\right)},\\ \Delta d_{k,1}^{\left(2\right)} & =\Delta a_{k,M\times N}^{\left(2\right)}∔\Delta a_{k,1}^{\left(2\right)}.\; \end{array}\right. \end{array}$$

Modular subtraction of each element from its next adjacent element as
(23)$$\begin{array}{c} \left\{\begin{array}{cc} \Delta d_{k,M\times N}^{\left(2\right)}\dot{-}\Delta d_{k,M\times N-1}^{\left(2\right)} & =\Delta a_{k,M\times N}^{\left(2\right)},\\ \Delta d_{k,M\times N-1}^{\left(2\right)}\dot{-}\Delta d_{k,M\times N-2}^{\left(2\right)} & =\Delta a_{k,M\times N-1}^{\left(2\right)},\\ \Delta d_{k,M\times N-2}^{\left(2\right)}\dot{-}\Delta d_{k,M\times N-3}^{\left(2\right)} & =\Delta a_{k,M\times N-2}^{\left(2\right)},\\ \; & \vdots \\ \Delta d_{k,2}^{\left(2\right)}\dot{-}\Delta d_{k,1}^{\left(2\right)} & =\Delta a_{k,2}^{\left(2\right)},\\ \Delta d_{k,1}^{\left(2\right)}∔\Delta d_{k,M\times N-1}^{\left(2\right)}\dot{-}\Delta d_{k,M\times N}^{\left(2\right)} & =\Delta a_{k,1}^{\left(2\right)}. \end{array}\right. \end{array}$$

Input $P_{0}$ and the plaintext image set $P_{1},P_{2},\ldots ,P_{M\times N}$ into the encryption algorithm in turn, and obtain the corresponding ciphertext image $C_{0}$ and $C_{1},C_{2},\ldots ,C_{M\times N}$ after two-round encryption, where $D_{0}^{\left(2\right)}=C_{0},D_{1}^{\left(2\right)}=C_{1},\ldots ,D_{M\times N}^{\left(2\right)}=C_{M\times N}$. Perform modular subtraction operation on each pixel value in $D_{1}^{\left(2\right)},D_{2}^{\left(2\right)},\ldots ,D_{M\times N}^{\left(2\right)}$ from each pixel value in $D_{0}^{\left(2\right)}$ to obtain $\Delta D_{1}^{\left(2\right)},\Delta D_{2}^{\left(2\right)},\Delta D_{3}^{\left(2\right)},\ldots ,\Delta D_{M\times N}^{\left(2\right)}$. According to Equation (Equation 23), $\Delta A_{1}^{\left(2\right)},\Delta A_{2}^{\left(2\right)},\Delta A_{3}^{\left(2\right)},\ldots ,\Delta A_{M\times N}^{\left(2\right)}$ are obtained, then the sum of all elements of the above matrices can be calculated as $\sum_{j=1}^{M\times N}\Delta a_{1,j}^{\left(2\right)},\sum_{j=1}^{M\times N}\Delta a_{2,j}^{\left(2\right)},\ldots ,\sum_{j=1}^{M\times N}\Delta a_{M\times N,j}^{\left(2\right)}$.

Step 3: Because the permutation operation does not change the sum of the element values in the matrix, so $\sum_{j=1}^{M\times N}\Delta d_{1,j}^{\left(1\right)}=\sum_{j=1}^{M\times N}\Delta a_{1,j}^{\left(2\right)},\sum_{j=1}^{M\times N}\Delta d_{2,j}^{\left(1\right)}=\sum_{j=1}^{M\times N}\Delta a_{2,j}^{\left(2\right)},\ldots ,\sum_{j=1}^{M\times N}\Delta d_{M\times N,j}^{\left(1\right)}=\sum_{j=1}^{M\times N}\Delta a_{M\times N,j}^{\left(2\right)}$.

According to Equation (Equation 20), one can obtain
(24)$$\begin{array}{c} \left\{\begin{array}{cc} \Delta d_{k,1}^{\left(1\right)} & =\Delta a_{k,M\times N}^{\left(1\right)}∔\Delta a_{k,1}^{\left(1\right)},\\ \Delta d_{k,2}^{\left(1\right)} & =\Delta a_{k,M\times N}^{\left(1\right)}∔\Delta a_{k,1}^{\left(1\right)}∔\Delta a_{k,2}^{\left(1\right)},\\ \; & \vdots \\ \Delta d_{k,j}^{\left(1\right)} & =\Delta a_{k,M\times N}^{\left(1\right)}∔\Delta a_{k,1}^{\left(1\right)}∔\Delta a_{k,2}^{\left(1\right)}∔\ldots ∔\Delta a_{k,j}^{\left(1\right)},\\ \; & \vdots \\ \Delta d_{k,M\times N}^{\left(1\right)} & =\Delta a_{k,M\times N}^{\left(1\right)}∔\Delta a_{k,1}^{\left(1\right)}∔\Delta a_{k,2}^{\left(1\right)}∔\ldots ∔\Delta a_{k,M\times N-2}^{\left(1\right)}∔\Delta a_{k,M\times N-1}^{\left(1\right)}∔\Delta a_{k,M\times N}^{\left(1\right)}. \end{array}\right. \end{array}$$

From Equation (Equation 21), it can be seen that $\Delta a_{k,j}^{\left(1\right)}=f_{kp_{h}}\left(\Delta p_{k,h}^{\left(1\right)}\right)=f_{kp_{h}}\left(\Delta p_{k,h}\right)$ because $\Delta p_{k,h}=p_{k,h}\dot{-}p_{0,h}=p_{k,h}$. From the properties of the plaintext image $P_{0}$ and the constructed plaintext image set $\left\{P_{1},P_{2},\ldots ,P_{M\times N}\right\}$, one has
(25)$$\begin{array}{c} \Delta p_{k,h}=\left\{\begin{array}{cc} 1 & \;\mathrm{if}\;h=k,\\ 0 & \;\mathrm{if}\;h\neq k, \end{array}\right. \end{array}$$
where k=1,2,…,M×N.

Because *h* and $kp_{h}$ are a permutation pair, when h=k, *k* and $kp_{k}$ are a permutation pair. That is, if $p_{k,k}$ is permuted by $a_{k,kp_{k}}^{\left(1\right)}$, $a_{k,kp_{k}}^{\left(1\right)}=p_{k,k}=1$, then there is
$$\Delta a_{k,j}^{\left(1\right)}=\left\{\begin{array}{cc} 1 & j=kp_{k},\\ 0 & j\neq kp_{k}, \end{array}\right.$$
where $kp_{k}=1,2,\ldots ,M\times N$, which is substituted into Equation (Equation 24), one can obtain
(26)$$\begin{array}{c} kp_{k}=\left\{\begin{array}{cc} 1 & \;\mathrm{if}\;\sum_{j=1}^{M\times N}\Delta a_{k,j}^{\left(2\right)}=M\times N,\\ M\times N & \;\mathrm{if}\;\sum_{j=1}^{M\times N}\Delta a_{k,j}^{\left(2\right)}=M\times N+1,\\ M\times N+1-\sum_{j=1}^{M\times N}\Delta a_{k,j}^{\left(2\right)} & \;\mathrm{if}\;\sum_{j=1}^{M\times N}\Delta a_{k,j}^{\left(2\right)}\neq M\times N\mathrm{and}\sum_{j=1}^{M\times N}\Delta a_{k,j}^{\left(2\right)}\neq M\times N+1. \end{array}\right. \end{array}$$

According to $\sum_{j=1}^{M\times N}\Delta a_{1,j}^{\left(2\right)},\sum_{j=1}^{M\times N}\Delta a_{2,j}^{\left(2\right)},\ldots ,\sum_{j=1}^{M\times N}\Delta a_{M\times N,j}^{\left(2\right)}$ obtained in the previous step, the equivalent permutation key KP used for position permutation can be obtained from the above equation.

Step 4: If the image to be cracked is *C*, $C_{0}$ is the ciphertext image corresponding to the all-zero plaintext. Let $\Delta C=C\dot{-}C_{0}$, then $\Delta D^{\left(2\right)}=\Delta C$. According to Equation (Equation 23), the adjacent two elements are modular subtracted to obtain $\Delta A^{\left(2\right)}$, and the equivalent permutation key KP is used to obtain $\Delta D^{\left(1\right)}$. Similarly, $\Delta A^{\left(1\right)}$ and $\Delta P=\Delta P^{\left(1\right)}$ can be obtained. Because $P_{0}$ is all-zero plaintext, so the deciphered plaintext is $P=P\dot{-}P_{0}=\Delta P$.

### 4.4. Security Analysis of Multi-Round Encryption

The chosen-ciphertext attack method was proposed in [38], which can crack the diffusion operation with ciphertext feedback and different permutation matrices in each round. The applicability of this method was summarized and demonstrated in detail in [40,41]. However, the above literature mainly gave this method for the case without feedback, and then extended it to the case with feedback directly. Through the detailed derivation of the encryption process, the steps of the attack method to crack the case with ciphertext feedback are given in this section. It is not only helpful for understanding the attack method, but also has a good inspiration for guiding the improvement of the algorithm. It should be pointed out that, if there is no special description for multi-round analysis, the symbol definitions given above are still used.

Let $a_{i,j}^{\left(t\right)}=f_{j}\left(p_{i,kp_{j}^{-1}}^{\left(t\right)}\right),j=1,2,\ldots ,M\times N$; this means that $a_{i,j}^{\left(t\right)}$ is the *j*-th element of the *i*-th permutation image in raster scan order in the *t*-th round of encryption is permutated from $p_{i,kp_{j}^{-1}}^{\left(t\right)}$ which is the $kp_{j}^{-1}$-th element of the *i*-th plaintext image $P_{i}^{\left(t\right)}$ in raster scan order in the *t*-th round of encryption; *j* and $kp_{j}^{-1}$ are a permutation pair. The encryption process of the original encryption algorithm can be expressed as a general model as
(27)$$\begin{array}{c} \left\{\begin{array}{cc} d_{i,j}^{\left(t\right)} & =f_{M\times N}\left(p_{i,kp_{M\times N}^{-1}}^{\left(t\right)}\right)∔\overset{j}{\underset{u=1}{\dot{\sum }}}f_{u}\left(p_{i,kp_{u}^{-1}}^{\left(t\right)}\right)∔\overset{j}{\underset{u=1}{\dot{\sum }}}{\hat{y}}_{u}^{\left(t\right)},\\ p_{i,j}^{\left(t\right)} & =d_{i,j}^{\left(t-1\right)}, \end{array}\right. \end{array}$$
where i=0,1,2,⋯, $j\in \left\{1,2,\ldots ,M\times N\right\}$, t=1,2,…,n, ∔ denotes modular addition operation, and $\dot{\sum }$ denotes summation operation of modular addition. As for $u\in \left\{M\times N,1,2,\ldots ,j\right\}$, *u* and $kp_{u}^{-1}$ are a permutation pair.

According to Equation (Equation 27), one has
(28)$$\begin{array}{cc} \; & p_{k,j}^{\left(t\right)}\dot{-}p_{l,j}^{\left(t\right)}=\Delta p_{k-l,j}^{\left(t\right)}=\Delta d_{k-l,j}^{\left(t-1\right)}\\ \; & =f_{M\times N}\left(\Delta p_{k-l,kp_{M\times N}^{-1}}^{\left(t-1\right)}\right)∔\overset{j}{\underset{u=1}{\dot{\sum }}}f_{u}\left(\Delta p_{k-l,kp_{u}^{-1}}^{\left(t-1\right)}\right). \end{array}$$

According to Equations (Equation 21) and (Equation 28), $\Delta a_{k-l,r}^{\left(t\right)}=a_{k,r}^{\left(t\right)}\dot{-}a_{l,r}^{\left(t\right)}$ is the difference between the *r*-th element of the permutation images $A_{k}^{\left(t\right)}$ and $A_{l}^{\left(t\right)}$ in *t*-th round of encryption. That is,
(29)$$\begin{array}{cc} \Delta a_{k-l,r}^{\left(t\right)} & =f_{r}\left(\Delta p_{k-l,kp_{r}^{-1}}^{\left(t\right)}\right)\\ \; & =f_{r}\left(f_{M\times N}\left(\Delta p_{k-l,kp_{M\times N}^{-1}}^{\left(t-1\right)}\right)∔\overset{kp_{r}^{-1}}{\underset{u=1}{\dot{\sum }}}f_{u}\left(\Delta p_{k-l,kp_{u}^{-1}}^{\left(t-1\right)}\right)\right), \end{array}$$
where k=1,2,⋯, l=0,1,2,⋯, r=1,2,…,M×N, t=1,2,…,n. By the way, *r* and $kp_{r}^{-1}$ are a permutation pair. As for $u\in \left\{M\times N,1,2,\ldots ,kp_{r}^{-1}\right\}$, *u* and $kp_{u}^{-1}$ are a permutation pair.

According to Equation (Equation 20), the difference from first to M×N-th element between diffusion image $D_{k}^{\left(t\right)}$ and $D_{0}^{\left(t\right)}$ (that is l=0) in the *t*-th round of encryption is:(30)$$\begin{array}{c} \left\{\begin{array}{cc} \Delta d_{k,1}^{\left(t\right)} & =\Delta a_{k,M\times N}^{\left(t\right)}∔\Delta a_{k,1}^{\left(t\right)},\\ \Delta d_{k,2}^{\left(t\right)} & =\Delta a_{k,M\times N}^{\left(t\right)}∔\Delta a_{k,1}^{\left(t\right)}∔\Delta a_{k,2}^{\left(t\right)},\\ \; & \vdots \\ \Delta d_{k,M\times N}^{\left(t\right)} & =\Delta a_{k,M\times N}^{\left(t\right)}∔\overset{M\times N}{\underset{v=1}{\dot{\sum }}}\Delta a_{k,v}^{\left(t\right)}. \end{array}\right. \end{array}$$

From Equation (Equation 28)–(Equation 30), one can further obtain
$$\begin{array}{cc} \Delta d_{k,1}^{\left(t\right)}= & f_{M\times N}\left(f_{M\times N}\left(\Delta p_{k,kp_{M\times N}^{-1}}^{\left(t-1\right)}\right)∔\overset{kp_{M\times N}^{-1}}{\underset{u=1}{\dot{\sum }}}f_{u}\left(\Delta p_{k,kp_{u}^{-1}}^{\left(t-1\right)}\right)\right)\\ \; & ∔f_{1}\left(f_{M\times N}\left(\Delta p_{k,kp_{M\times N}^{-1}}^{\left(t-1\right)}\right)∔\overset{kp_{1}^{-1}}{\underset{u=1}{\dot{\sum }}}f_{u}\left(\Delta p_{k,kp_{u}^{-1}}^{\left(t-1\right)}\right)\right)\\ \Delta d_{k,2}^{\left(t\right)}= & f_{M\times N}\left(f_{M\times N}\left(\Delta p_{k,kp_{M\times N}^{-1}}^{\left(t-1\right)}\right)∔\overset{kp_{M\times N}^{-1}}{\underset{u=1}{\dot{\sum }}}f_{u}\left(\Delta p_{k,kp_{u}^{-1}}^{\left(t-1\right)}\right)\right)\\ \; & ∔f_{1}\left(f_{M\times N}\left(\Delta p_{k,kp_{M\times N}^{-1}}^{\left(t-1\right)}\right)∔\overset{kp_{1}^{-1}}{\underset{u=1}{\dot{\sum }}}f_{u}\left(\Delta p_{k,kp_{u}^{-1}}^{\left(t-1\right)}\right)\right)\\ \; & ∔f_{2}\left(f_{M\times N}\left(\Delta p_{k,kp_{M\times N}^{-1}}^{\left(t-1\right)}\right)∔\overset{kp_{2}^{-1}}{\underset{u=1}{\dot{\sum }}}f_{u}\left(\Delta p_{k,kp_{u}^{-1}}^{\left(t-1\right)}\right)\right)\\ \; & \vdots \\ \Delta d_{k,M\times N}^{\left(t\right)}= & f_{M\times N}\left(f_{M\times N}\left(\Delta p_{k,kp_{M\times N}^{-1}}^{\left(t-1\right)}\right)∔\overset{kp_{M\times N}^{-1}}{\underset{u=1}{\dot{\sum }}}f_{u}\left(\Delta p_{k,kp_{u}^{-1}}^{\left(t-1\right)}\right)\right)\\ \; & ∔\overset{M\times N}{\underset{v=1}{\dot{\sum }}}f_{v}\left(f_{M\times N}\left(\Delta p_{k,kp_{M\times N}^{-1}}^{\left(t-1\right)}\right)∔\overset{kp_{v}^{-1}}{\underset{u=1}{\dot{\sum }}}f_{u}\left(\Delta p_{k,kp_{u}^{-1}}^{\left(t-1\right)}\right)\right) \end{array}$$

Since the modular addition and permutation can be processed out of order and ended up with the same result, after t=n rounds of encryption, the pixel difference result $\Delta d_{k,j}^{\left(n\right)}\left(j=1,2,\ldots ,M\times N\right)$ of diffusion images $D_{k}^{\left(n\right)}$, and $D_{0}^{\left(n\right)}$ can be expressed as the linear combination of the difference pixel point $\Delta p_{k,j}^{\left(n-1\right)}\left(j=1,2,\ldots ,M\times N\right)$ of n−1-round plaintext $P_{k}^{\left(n-1\right)}$ and $P_{0}^{\left(n-1\right)}$ modulo 256. $\Delta d_{k,j}^{\left(n\right)},j=1,2,\ldots ,M\times N$ can be recursively expressed as the linear combination of $\Delta p_{k,j}^{\left(1\right)}=\Delta p_{k,j},j=1,2,\ldots ,M\times N$ modulo 256, which is
(31)$$\begin{array}{c} \left[\begin{array}{c} \Delta d_{k,1}^{\left(n\right)}\\ \Delta d_{k,2}^{\left(n\right)}\\ \vdots \\ \Delta d_{k,M\times N}^{\left(n\right)} \end{array}\right]=\left(\left[\begin{array}{cccc} b_{11} & b_{11} & \cdots & b_{1M\times N}\\ b_{21} & b_{22} & \cdots & b_{2M\times N}\\ \vdots & \vdots & \cdots & \vdots \\ b_{M\times N,1} & b_{M\times N,2} & \cdots & b_{M\times N,M\times N} \end{array}\right]\times \left[\begin{array}{c} \Delta p_{k,1}\\ \Delta p_{k,2}\\ \vdots \\ \Delta p_{k,M\times N} \end{array}\right]\right)\mathrm{Mod}256, \end{array}$$
where Mod represents the modulo of each component of the vector.

For an encryption system, any plaintext image must have only one corresponding ciphertext image. At the same time, any ciphertext image can only be decrypted to one plaintext image; otherwise, the encryption algorithm will not be established. In addition, the number of pixels in the plaintext image and the ciphertext image is constant, so the coefficient matrix is a square matrix, and its rank must be M×N. Furthermore, for the *n*-round encryption algorithm, the coefficient matrix is represented by the symbol
(32)$$\begin{array}{c} B=\left[\begin{array}{cccc} b_{11} & b_{11} & \cdots & b_{1M\times N}\\ b_{21} & b_{22} & \cdots & b_{2M\times N}\\ \vdots & \vdots & \cdots & \vdots \\ b_{M\times N,1} & b_{M\times N,2} & \cdots & b_{M\times N,M\times N} \end{array}\right]. \end{array}$$

$\Delta \alpha_{k}={\left[\Delta d_{k,1}^{\left(n\right)},\Delta d_{k,2}^{\left(n\right)},\cdots ,\Delta d_{k,M\times N}^{\left(n\right)}\right]}^{T}$ represents the one-dimensional vector converted from the difference matrix between the ciphertext images $D_{k}^{\left(n\right)}$ and $D_{0}^{\left(n\right)}$ in the *n*-th round of encryption according to the raster scan order. In addition, $\Delta \beta_{k}={\left[\Delta p_{k,1},\Delta p_{k,2},\cdots ,\Delta p_{k,M\times N}\right]}^{T}$ represents the one-dimensional vector converted from the difference matrix between the plaintext $P_{k}^{\left(1\right)}$ corresponding to the ciphertext image $D_{k}^{\left(n\right)}$ and the plaintext $P_{0}^{\left(1\right)}$ corresponding to the ciphertext image $D_{0}^{\left(n\right)}$, in the *n*-th round of encryption, according to the raster scan order. Then, the above equation can be expressed as
(33)$$\Delta \alpha_{k}=\left(B\times \Delta \beta_{k}\right)\mathrm{Mod}256.$$

Consider a set of standard orthogonal bases $e_{1},e_{2},\ldots ,e_{M\times N}$, where $e_{1}={\left[1,0,\ldots ,0\right]}^{T}$, $e_{2}={\left[0,1,0,\ldots ,0\right]}^{T},\ldots ,e_{M\times N}={\left[0,\ldots ,0,1\right]}^{T}$. Then, any one-dimensional vector $\Delta \alpha ={\left[c_{1},c_{2},\ldots ,c_{M\times N}\right]}^{T}$ can be expressed as
(34)$$\Delta \alpha =\left(c_{1}\times e_{1}+c_{2}\times e_{2}+\ldots +c_{M\times N}\times e_{M\times N}\right)\mathrm{Mod}256.$$

According to Equation (Equation 33), $e_{1},e_{2},\ldots ,e_{M\times N}$ corresponds to $\Delta \beta_{1},\Delta \beta_{2},\ldots ,\Delta \beta_{M\times N}$, so
(35)$$\begin{array}{c} \left\{\begin{array}{cc} e_{1} & =\left(B\times \Delta \beta_{1}\right)\mathrm{Mod}256,\\ e_{2} & =\left(B\times \Delta \beta_{2}\right)\mathrm{Mod}256,\\ \; & \vdots \\ e_{M\times N} & =\left(B\times \Delta \beta_{M\times N}\right)\mathrm{Mod}256, \end{array}\right. \end{array}$$
which is substituted into Equation (Equation 34) to obtain
(36)$$\begin{array}{cc} \Delta \alpha = & (c_{1}\times \left(B\times \Delta \beta_{1}\right)\mathrm{Mod}256+c_{2}\times \left(B\times \Delta \beta_{2}\right)\mathrm{Mod}256+\\ \; & \ldots +c_{M\times N}\times \left(B\times \Delta \beta_{M\times N}\right)\mathrm{Mod}256)\mathrm{Mod}256\\ = & \left(c_{1}\times B\times \Delta \beta_{1}+c_{2}\times B\times \Delta \beta_{2}+\ldots +c_{M\times N}\times B\times \Delta \beta_{M\times N}\right)\mathrm{Mod}256\\ = & \left(B\times \left(c_{1}\times \Delta \beta_{1}+c_{2}\times \Delta \beta_{2}+\ldots +c_{M\times N}\times \Delta \beta_{M\times N}\right)\right)\mathrm{Mod}256. \end{array}$$

According to Equation (Equation 33), Δβ corresponding to Δα is
(37)$$\Delta \beta =\left(c_{1}\times \Delta \beta_{1}+c_{2}\times \Delta \beta_{2}+\ldots +c_{M\times N}\times \Delta \beta_{M\times N}\right)\mathrm{Mod}256.$$

The flow chart for cracking the multi-round encryption is shown in Figure 5.

The details of our cracking process consist of four steps, as given below.

Step 1: Record a ciphertext image of M×N to be cracked as
$$C=\left[\begin{array}{cccc} c_{1} & c_{2} & \cdots & c_{N}\\ c_{N+1} & c_{N+2} & \cdots & c_{2\times N}\\ \vdots & \vdots & \cdots & \vdots \\ c_{\left(M-1\right)\times N+1} & b_{\left(M-1\right)\times N+2} & \cdots & b_{M\times N} \end{array}\right].$$

Firstly, an all-zero ciphertext image of M×N is denoted by $C_{0}$, then a ciphertext image set $\left\{C_{1},C_{2},\ldots ,C_{M\times N}\right\}$ is constructed, so that the *k*-th element in $C_{k}$ according to the raster scan order is set to 1, and the rest is 0 where $k\in \left\{1,2,\ldots ,M\times N\right\}$, as $C_{1}=\left[\begin{array}{cccc} 1 & 0 & \cdots & 0\\ 0 & 0 & \cdots & 0\\ \vdots & \vdots & & \vdots \\ 0 & 0 & \cdots & 0 \end{array}\right]$, $C_{2}=\left[\begin{array}{cccc} 0 & 1 & \cdots & 0\\ 0 & 0 & \cdots & 0\\ \vdots & \vdots & & \vdots \\ 0 & 0 & \cdots & 0 \end{array}\right]$, ⋯, $C_{M\times N}=\left[\begin{array}{cccc} 0 & 0 & \cdots & 0\\ 0 & 0 & \cdots & 0\\ \vdots & \vdots & & \vdots \\ 0 & 0 & \cdots & 1 \end{array}\right].$

Perform modular subtraction operations of $C_{0}$ from $C,C_{1},C_{2},\ldots ,C_{M\times N}$, respectively. Because $C_{0}$ is an all-zero ciphertext, the one-dimensional vector converted from $\Delta D^{\left(n\right)}=C,\Delta D_{1}^{\left(n\right)}=C_{1},\Delta D_{2}^{\left(n\right)}=C_{2},\ldots ,\Delta D_{M\times N}^{\left(n\right)}=C_{M\times N}$ according to the raster scan order is actually the aforementioned $\Delta \alpha ,e_{1},e_{2},\ldots ,e_{M\times N}$.

Input $C_{0},C_{1},C_{2},\ldots ,C_{M\times N}$ into the decryption machine to obtain a set of corresponding plaintext images, which are denoted as $P_{0},P_{1},\ldots ,P_{M\times N}$.

Step 2: Perform modular subtraction operations of $P_{0}$ from $P_{1},P_{2},\ldots ,P_{M\times N}$, respectively, and the result $\Delta P_{1}=P_{1}\dot{-}P_{0},\Delta P_{2}=P_{2}\dot{-}P_{0},\ldots ,\Delta P_{M\times N}=P_{M\times N}\dot{-}P_{0}$ is converted into a one-dimensional vector according to the raster scan order, which is actually the aforementioned $\Delta \beta_{1},\Delta \beta_{2},\ldots ,\Delta \beta_{M\times N}$.

Step 3: Therefore, Equation (Equation 37) can also be written as
(38)$$\Delta P=c_{1}\times \Delta P_{1}∔c_{2}\times \Delta P_{2}∔\ldots ∔c_{M\times N}\times \Delta P_{M\times N}.$$

Step 4: By $\Delta P=\Delta P^{\left(1\right)}=P\dot{-}P_{0}$, the plaintext is finally obtained as
(39)$$P=\Delta P∔P_{0}=\left(c_{1}\times \Delta P_{1}∔c_{2}\times \Delta P_{2}∔\ldots ∔c_{M\times N}\times \Delta P_{M\times N}\right)∔P_{0}.$$

## 5. Numerical Simulation Experiment

The experimental environment is Intel(R) Core(TM) i5-3230M processor, CPU frequency of 2.60 GHz, 8.00 GB memory, Windows 10 operating system, and MATLABR2021a. The grayscale images Lena, Cameraman, Tiffany, Pepper, Baboon and Sailboat with the size of 128×128 are selected for single-round, two-round, and multi-round numerical simulation experiments. The key is selected as follows:$$\begin{array}{c} a_{0}=\left\{1111010010,0110101101,1110001101,1111110000\right\},\\ b_{0}=\left\{0001111100,1011100111,0010010010,1011000000\right\},\\ x=\{1010101100,1101011011,1110001110,1000100000\},\\ y=\{1111111101,1110100111,1101011111,1111001010\},\\ T=\{1101000000,0001011010,0011001100,1000011011\},\\ C_{1}=\left\{1011111100,0010000101\right\},C_{2}=\left\{1100001010,1111000110\right\}. \end{array}$$

### 5.1. Experimental Results

This paper analyzes the original algorithm in the case of single-round, two-round, and multi-round of encryption. Because the first two analysis methods belong to the chosen-plaintext attack and the third one belongs to the chosen-ciphertext attack, the first two analysis methods are stronger in terms of assumptions made and data requirements; they are only applicable to themselves, but the cracking speed is faster. The multi-round analysis method mentioned in this paper is applicable to any number of encryption rounds and has universality.

This paper verifies the original encryption algorithm and the analysis results by writing MATLAB programs. According to the original encryption algorithm and the cracking algorithm, the encryption and the decryption programs and cracking programs in single-round, two-round, and multi-round are written, respectively. The related experimental results are shown in Figure 6, Figure 7, Figure 8, Figure 9, Figure 10, Figure 11 and Figure 12.

Figure 6 and Figure 7 respectively list the intermediate experimental results, cracking results and relevant metrics of 128×128 Lena image and Cameraman image for single-round encryption. It can be seen that, after encryption, the histogram of ciphertext is uniform and cannot reflect plaintext information. The histogram of intermediate ciphertext obtained by cracking the equivalent diffusion key is the same as that of plaintext, indicating that the diffusion step is cracked. Then, through the obtained permutation equivalent key, the deciphered image is obtained. Compared with the original plaintext image, the deciphered image is exactly the same, which shows that the cracking algorithm is correct.

Figure 8 and Figure 9 respectively list the intermediate experimental results and cracking results of the Tiffany image and Pepper image of 128×128 size for two-round of encryption. A set of specially constructed chosen-plaintext is input into the encryption algorithm to obtain a set of plaintext–ciphertext pairs, and according to these plaintext–ciphertext pairs, the permutation matrix is finally cracked. Then, through the differential processing and the obtained permutation matrix, the plaintext is finally cracked, and the comparison between the deciphered image and the plaintext image is exactly the same, which shows that the cracking algorithm is correct.

Figure 10 and Figure 11 list the intermediate experimental results and cracking results of 128×128 size Baboon image and Sailboat image for multi-round encryption (without losing generation, the multi-round part is encrypted in three rounds). It can be seen that, by inputting a group of specially constructed chosen-ciphertext into the decryption algorithm, a group of ciphertext–plaintext pairs are obtained, and then the plaintext is finally cracked through differential processing. The comparison between the recovered image and the plaintext image shows that the cracking algorithm is correct.

Theoretically, for a ciphertext image of M×N size, using multi-round of a cracking algorithm to completely recover the ciphertext image requires the construction of M×N ciphertext images and an all-zero ciphertext, a total of M×N+1 ciphertext–plaintext pairs. Because it is difficult to obtain the permission of decryption machine in reality, it is of practical significance to reduce the number of ciphertext–plaintext pairs. Figure 12 shows the effect of constructing different number of ciphertext–plaintext pairs on the finally recovered plaintext image.

It can be seen that the plaintext can be recovered better without some ciphertext–plaintext pairs. In reality, the appropriate number of ciphertext–plaintext pairs can be obtained by constructing an appropriate number of ciphertext, which can reduce the data complexity and improve the cracking efficiency while meeting the cracking requirements.

### 5.2. Attack Complexity

The attack complexity of cryptanalysis generally includes time complexity and data complexity. However, the time complexity is affected by the performance of the computer and the written cracking program, so it is uncertain. For cryptanalysis, the most important thing is the data complexity, that is, the number of plaintext or ciphertext required to crack an encryption algorithm. The following will discuss the data complexity of the cracking algorithm for the case of single-round, two-round, and multi-round in case of complete cracking.

In the case of single-round, for the grayscale image of M×N, when the key is unknown, the plaintext-ciphertext pair required to decipher the equivalent diffusion key and the equivalent permutation key is $\left(1+\left\lceil \log_{256}\left(M\times N\right)\right\rceil \right)$, so the data complexity is $O\left(\log_{}\left(M\times N\right)\right)$.

In the case of two-round, for the grayscale image of M×N, when the key is unknown, the number of chosen-plaintexts required to decipher is $\left(1+M\times N\right)$, so the data complexity is $O\left(M\times N\right)$.

In the case of multi-round, for the grayscale image of M×N, when the key is unknown, the number of chosen-ciphertexts required to decipher is $\left(1+M\times N\right)$, so the data complexity is $O\left(M\times N\right)$.

### 5.3. Improvement Plan

From the analysis of this article, it can be seen that the original encryption algorithm has the following vulnerabilities and deficiencies.

(1) The decryption equation of diffusion operation is incorrect. The original decryption result is slightly different from the original encrypted image.

(2) The original encryption algorithm attempts to increase the nonlinear factors by using index matrices and adding a round-down operation to the diffusion equation, for improving the security of the algorithm. However, the analysis found that the above processes can not provide higher security. Instead, an additional permutation is added, resulting in increasing the encryption time. Through the corresponding processing and transformation, the algorithm is still linear and can not resist against the chosen-ciphertext attack.

(3) Under the differential attack, the original diffusion key is completely useless.

In view of the above shortcomings, the following improvement suggestions are put forward.

Cancel the permutation 2 operation. Add new operations in the diffusion process, such as adding S-box, to improve the security of the algorithm. As a nonlinear device, S-box can be applied to the original algorithm to solve its problem. The design of the S-box needs to satisfy cryptographic properties such as nonlinearity, strict avalanche criterion, algebraic immunity, differential uniformity, and correlation immunity [25]. The improvement is given below by taking the S-box as an example.

The permutation operation of the original encryption algorithm is retained. Cancel the permutation operation in the diffusion process of the original encryption algorithm, and complete the diffusion operation according to the original diffusion equation. Take the first 256 bits of the matrix *X* (if it is less than 256 bits, use the chaotic system to iterate the insufficient bits), obtain a 256-bit index matrix according to the original algorithm, and then form a matrix of 16×16 according to the raster scan method to obtain an S-box. Each pixel of the diffusion image is indexed into the S-box according to the first four bits and the last four bits, and the value of the original pixel is replaced with the corresponding value in the S-box. According to the above steps, the encryption algorithm is carried out for two or more rounds, and the decryption algorithm is the inverse operation of the encryption algorithm.

## 6. Conclusions

In this paper, the security analysis of the image encryption algorithm based on two-dimensional infinite collapse map is carried out, and the definitions and properties of the round-down operation, the fractional part operation of real numbers, and the modular operation are given. At the same time, by using these theorems, the error of the original diffusion equation is found out, and finally the original encryption algorithm is processed into a general permutation–diffusion structure, and the diffusion structure is processed into a modular addition of the ciphertext feedback and the element of diffusion matrix. On this basis, the single-round, two-round, and multi-round situations are analyzed and discussed respectively. It not only deepens the understanding of the original encryption process, but also helps to guide the improvement of the original encryption algorithm. The correctness of the analysis process is verified by experiments. Finally, the attack complexity is discussed, and suggestions for improvement are given to avoid the shortcomings of the original encryption algorithm.

## Figures and Tables

**Figure 1 entropy-24-01023-f001:**
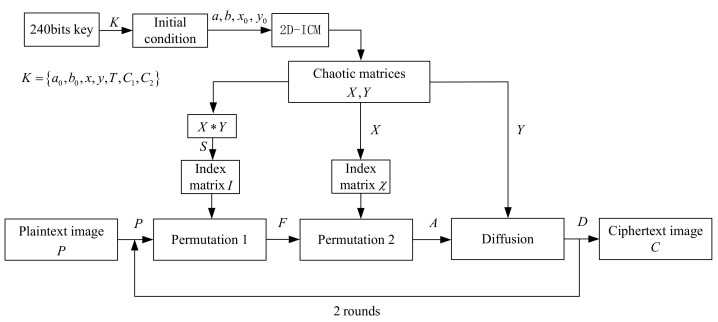
The algorithmic structure for ICMIE.

**Figure 2 entropy-24-01023-f002:**
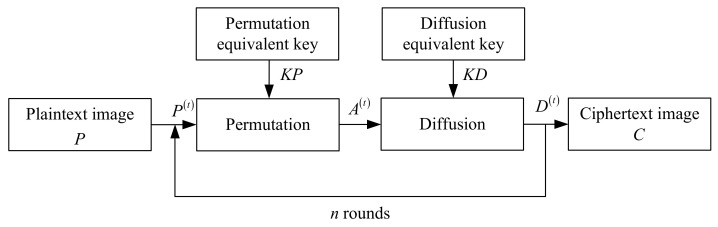
The block diagram of n-round chaotic image encryption.

**Figure 3 entropy-24-01023-f003:**
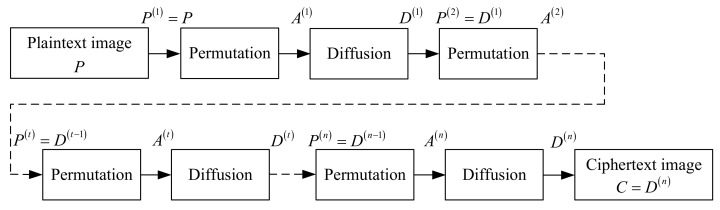
The block diagram of n-round chaotic image encryption without feedback.

**Figure 4 entropy-24-01023-f004:**
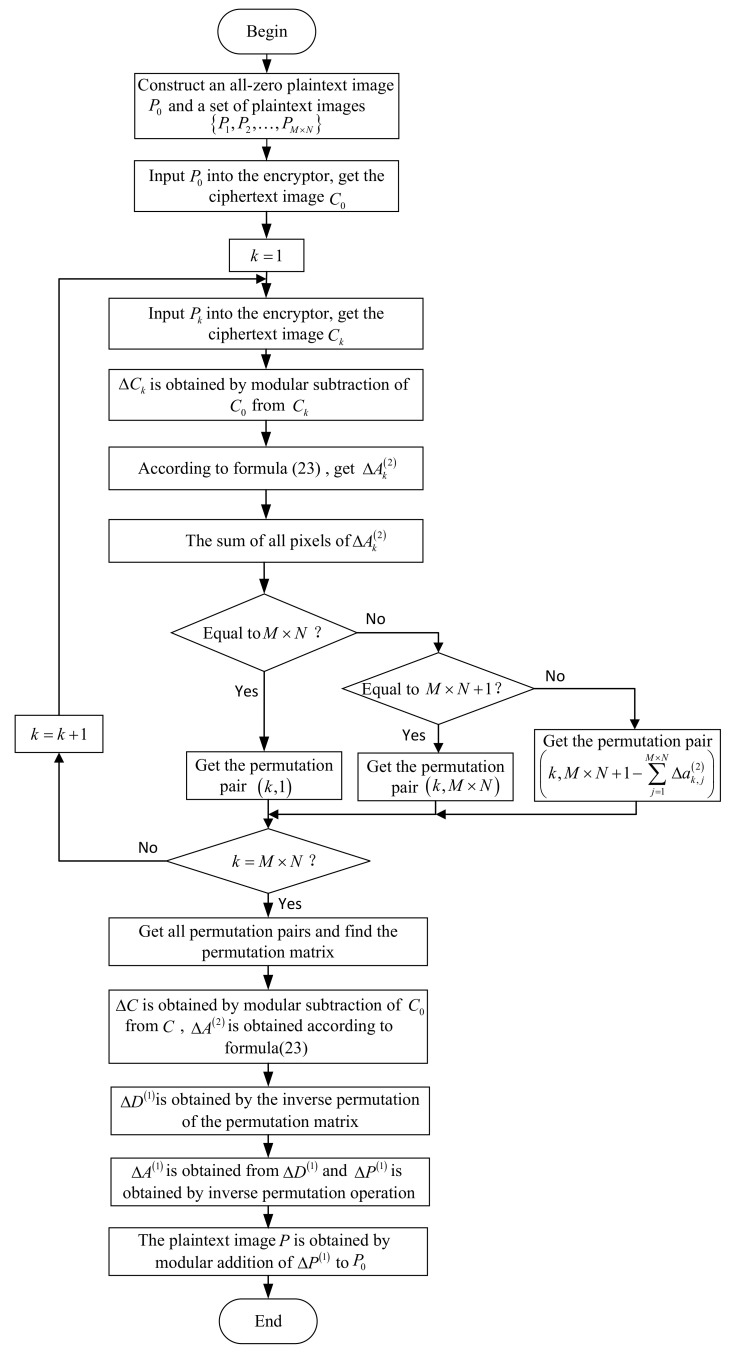
The flow chart for cracking the two-round encryption.

**Figure 5 entropy-24-01023-f005:**
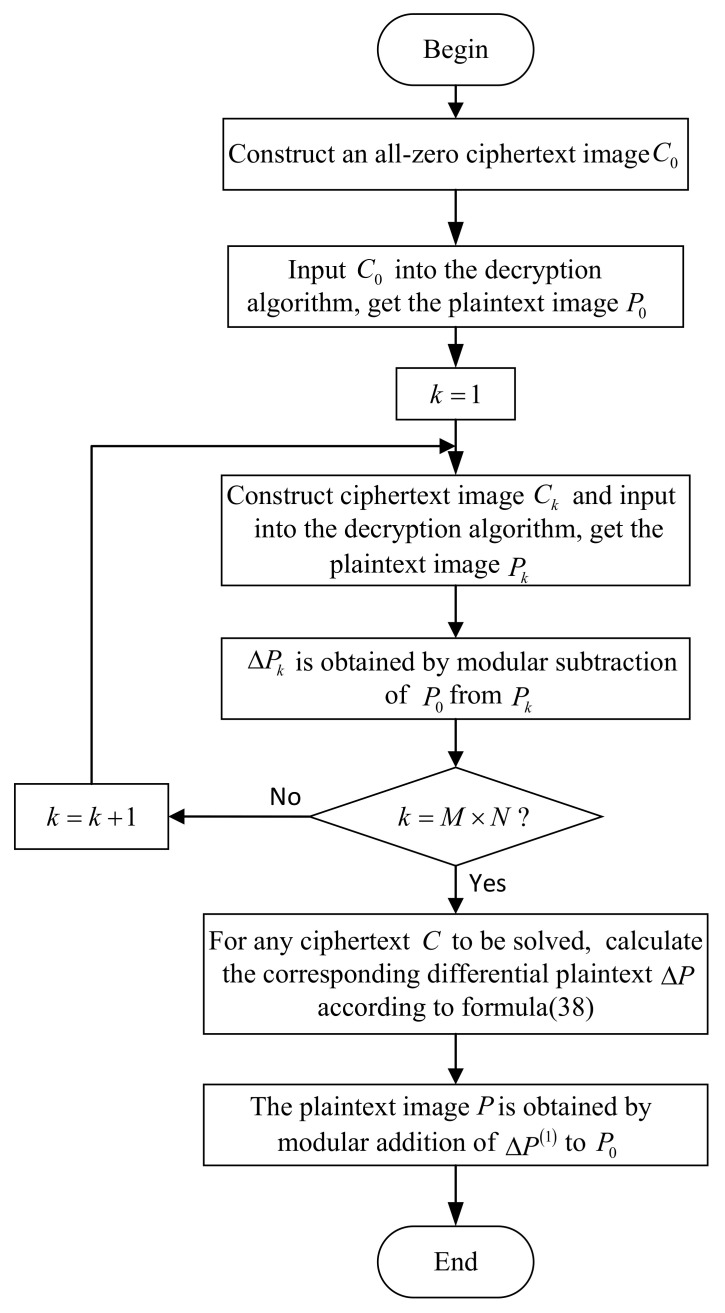
The flow chart for cracking the multi-round encryption.

**Figure 6 entropy-24-01023-f006:**
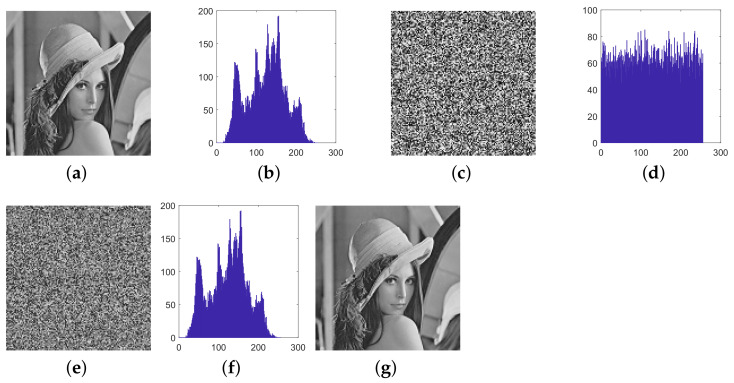
The experimental results of cracking the single-round encryption of Lena, (**a**) plaintext image of Lena; (**b**) histogram of plaintext Lena; (**c**) ciphertext of Lena; (**d**) histogram of ciphertext Lena; (**e**) inverse diffusion image of Lena; (**f**) histogram of (**e**); (**g**) the recovered image.

**Figure 7 entropy-24-01023-f007:**
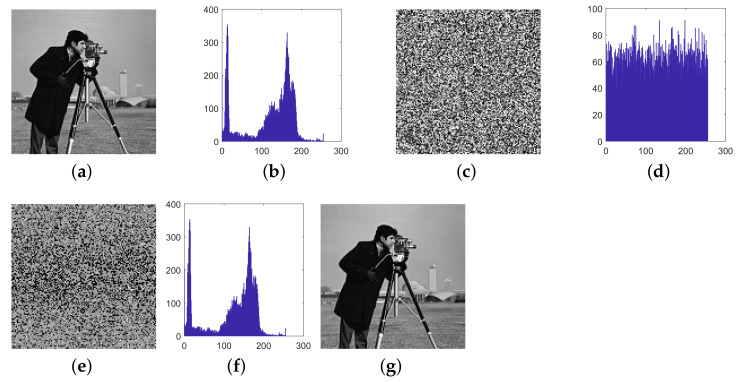
The experimental results of cracking the single-round encryption of Cameraman, (**a**) plaintext image of Cameraman; (**b**) histogram of plaintext Cameraman; (**c**) ciphertext of Cameraman; (**d**) histogram of ciphertext Cameraman; (**e**) inverse diffusion image of Cameraman; (**f**) histogram of (**e**); (**g**) the recovered image.

**Figure 8 entropy-24-01023-f008:**
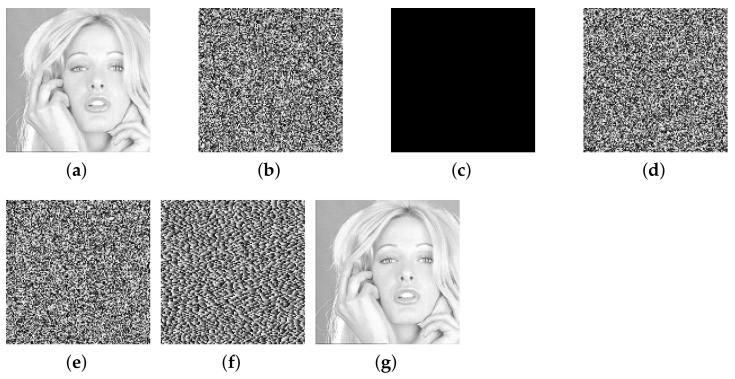
The cracking experiment results of two-round of encryption of Tiffany, (**a**) plaintext image of Tiffany; (**b**) ciphertext of Tiffany; (**c**) all-zero chosen-plaintext $P_{0}$; (**d**) the corresponding ciphertext of (**c**); (**e**) calculated differential image ΔC; (**f**) intermediate cracking results $\Delta D^{\left(1\right)}$; (**g**) the recovered image.

**Figure 9 entropy-24-01023-f009:**
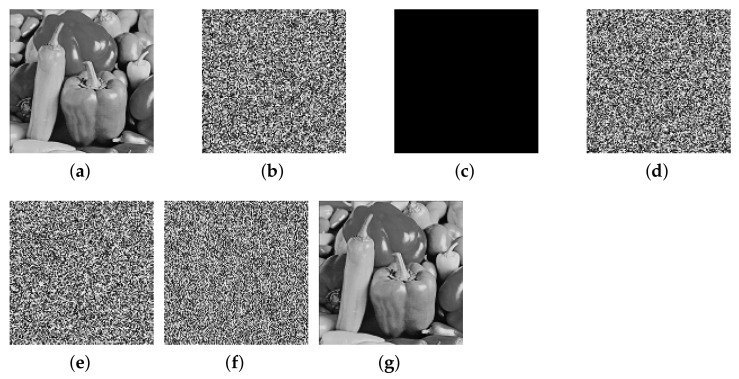
The cracking experiment results of two-round of encryption of Pepper, (**a**) plaintext image of Pepper; (**b**) ciphertext of Pepper; (**c**) all-zero chosen-plaintext $P_{0}$; (**d**) the corresponding ciphertext of (**c**); (**e**) calculated differential image ΔC; (**f**) intermediate cracking results $\Delta D^{\left(1\right)}$; (**g**) the recovered image.

**Figure 10 entropy-24-01023-f010:**
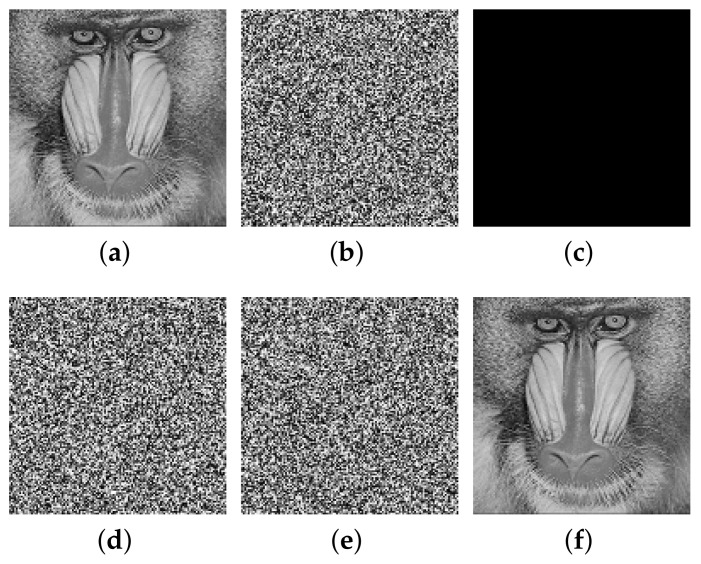
The cracking experiment results of multi-round of encryption of Baboon, (**a**) plaintext image of Baboon; (**b**) ciphertext of Baboon; (**c**) all-zero chosen-ciphertext $C_{0}$; (**d**) the corresponding plaintext of (**c**); (**e**) intermediate cracking results ΔP; (**f**) the recovered image.

**Figure 11 entropy-24-01023-f011:**
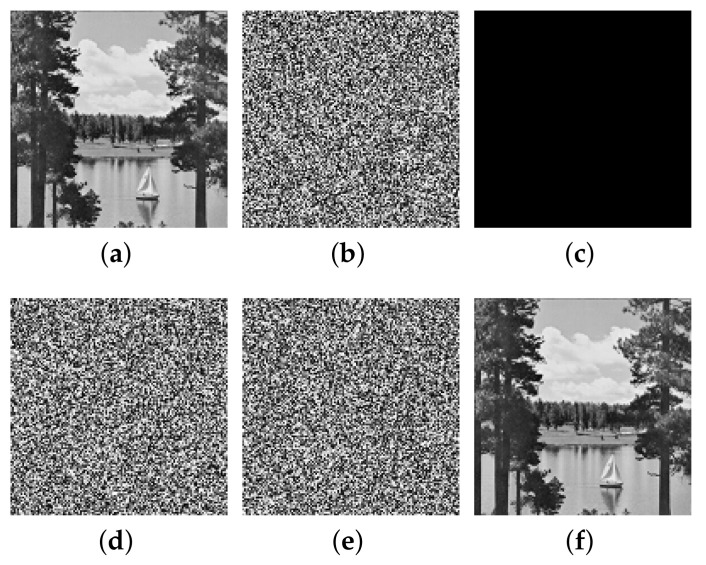
The cracking experiment results of multi-round of encryption of Sailboat, (**a**) plaintext image of Sailboat; (**b**) ciphertext of Sailboat; (**c**) all-zero chosen-ciphertext $C_{0}$; (**d**) the corresponding plaintext of (**c**); (**e**) intermediate cracking results ΔP; (**f**) the recovered image.

**Figure 12 entropy-24-01023-f012:**
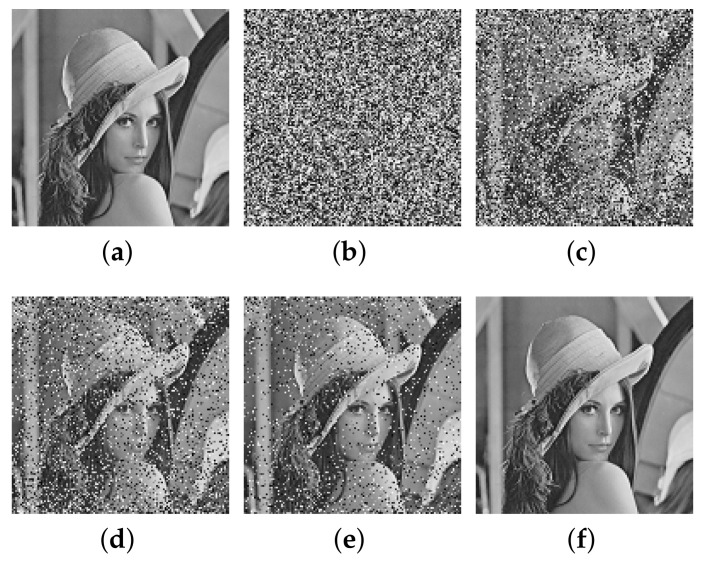
The cracking effect of partial ciphertext–plaintext pairs on multi-round cracking algorithm, (**a**) plaintext image of Lena; (**b**) recovered image with 50% ciphertext–plaintext pairs; (**c**) recovered image with 80% ciphertext–plaintext pairs; (**d**) recovered image with 90% ciphertext–plaintext pairs; (**e**) recovered image with 95% ciphertext–plaintext pairs; (**f**) recovered image with 100% ciphertext–plaintext pairs.

## Data Availability

Not applicable.

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
