# Peer review of "Security Analysis of the Image Encryption Algorithm Based on a Two-Dimensional Infinite Collapse Map"

_entropy, 2022, doi:10.3390/e24081023_

Round 1
Reviewer 1 Report
1. What is the advantage of using this methodology? In the literature, there are several algorithms they already give very promising results even after attacks.
2. Authors gave all details of the theorems and scientific background. It is fine, but it should be supported with an explanation of all these theories to get readers' interest.
3. Experimental part is weak. The expectation is the using several metrics and images, and also a comparison with other methods.
4. How about attacks? How your method is resisting the attacks? Do you have any results to show that?
Round 2
Reviewer 1 Report
The authors answered all questions and applied the necessary changes to the manuscript. The proposed work is novel, interesting, and good quality of writing. Experimental results are promising.
Reviewer 2 Report
Page 20, line 332: to completely recovery should be 'to completely recover'